# Improving machine classification using human uncertainty measurements

## Abstract

As deep CNN classifier performance using ground-truth labels has begun to asymptote at near-perfect levels, a key aim for the field is to extend training paradigms to capture further useful structure in natural image data and improve model robustness and generalization. In this paper, we present a novel natural image benchmark for making this extension, which we call `CIFAR10H`. This new dataset comprises a human-derived, full distribution over labels for each image of the `CIFAR10` test set, offering the ability to assess the generalization of state-of-the-art `CIFAR10` models, as well as investigate the effects of including this information in model training. We show that classification models trained on `CIFAR10` do not generalize as well to our dataset as it does to traditional extensions, and that models fine-tuned using our label information are able to generalize better to related datasets, complement popular data augmentation schemes, and provide robustness to adversarial attacks. We explain these improvements in terms of better empirical approximations to the expected loss function over natural images and their categories in the visual world.

## 1 Introduction

On natural-image classification benchmarks, state-of-the-art convolutional neural network (CNN) models have been said to equal or even surpass human performance, measured in terms of the accuracy of a model's top category choice for a test set of held-out images. As accuracy gains have begun to asymptote at near-perfect levels (Gastaldi, 2017), there has been increasing focus on out-of-training-set performance—in particular, the ability to generalize to related stimuli (Recht et al., 2018), and robustness to adversarial examples (Kurakin et al., 2016). On these tasks, by contrast, CNNs tend to perform rather poorly, whereas humans continue to perform well.

These diverging properties are likely closely related; by focusing on the match between a model's top output and a single ground-truth category, we are likely overfitting to these benchmarks to the detriment of generalization and robustness. This observation is supported by emerging empirical tests of near-in-sample generalization (Recht et al., 2018), as well as results from the theoretical literature that under current training paradigms CNNs could memorize training examples rather than learn to generalize from them (Zhang et al., 2017). If we have saturated accuracy as the primary measure of classification performance, a critical question is what we should turn to next.

One promising candidate is the loss between a model's distribution over labels $y$ for any image, $x$, and the natural distribution of images and their categories, $c$, in the world, $p(c|x)$; a probabilistic extension of top-1 accuracy to top-$n$. The uncertainty over labels inherent in the underlying data distribution is a rich source of information that can be leveraged not only for explaining how confident models should be when they are right, but for learning relevant structure in the labels—for example, cross-category confusions—that can reduce loss when they are wrong. Indeed, there is often a lack of human consensus on the category of an object, and the way the categorization decisions deviate from the ground truth often conveys important information about the structure of the visual world (Lakoff, 2008). This information can potentially be leveraged to discover more generalizable and more robust perceptual boundaries.

In this paper, we present a novel image database that can support this new objective, which we call `CIFAR10H`. This database comprises over 500k human classifications over the test set of the `CIFAR10` natural image dataset (Krizhevsky, 2009), which allows us to train and evaluate models

on the full probability distribution across categories for all images. We assess the generalization of pretrained CNN classification models on this new dataset, and show that when such models are fine-tuned using the full distribution of human labels, their generalization abilities improve. Our key theoretical contribution is to show that sets of human image classification decisions for an image can be used as a suitable proxy for the true distribution over labels, and that this is an efficient strategy for capturing distributional assumptions during training that confer better generalization and robustness.

## 2 EMPIRICAL RISK MINIMIZATION BEYOND THE MODE

In statistical learning, our goal is to learn the best model of an underlying data distribution over a set of random variables, for example, features $X$ and labels $Y$. In general, we are given a family of models $f$, and are tasked to find a member of that family, indexed by parameters $\theta$, that minimizes the expected loss

$$\arg\min_\theta \int \mathcal{L}(f_\theta, x, y)\, p(x, y)\, dx\, dy, \tag{1}$$

where $\mathcal{L}$ is a loss function that penalizes deviations of model predictions from the data distribution, and $p(x, y)$ is the data distribution itself. Since in general, we do not know $p(x, y)$, we approximate it by the empirical distribution over a set of $n$ samples $(x_1, y_1), \cdots, (x_n, y_n)$

$$\arg\min_\theta \int \mathcal{L}(f_\theta, x, y)\, p(x, y)\, dx\, dy \approx \arg\min_\theta \int \mathcal{L}(f_\theta, x, y)\, p_\delta(x, y)\, dx\, dy, \tag{2}$$

where $p_\delta(x, y)$ is an empirical distribution defined on the set of samples as follows

$$p_\delta(x, y) = \frac{1}{n} \sum_{i=1}^n \delta(x = x_i, y = y_i) \tag{3}$$

where $\delta(x = x_i, y = y_i)$ represents a Dirac mass centered at $(x_i, y_i)$. We can thus define the empirical risk as

$$\sum_{i=1}^n \mathcal{L}(f_\theta, x_i, y_i). \tag{4}$$

In supervised learning, we use a training subset of samples $(x_i, y_i)$ to learn parameters $\theta$ that minimize the loss between model outputs $f_\theta(x_i)$, and the "true" outputs $y_i$ associated with them. For natural image classification, our family of functions $f$ are CNNs parameterized by weights $\theta$, our inputs $x$ are vectors of pixel intensities, and our targets $y$ are category labels $c$ corresponding to each image. The probabilistic interpretation of these targets is that of samples drawn from the true $p(y|x)$. For a given image $x_i$ generated from its true category $c_t$ in a set of $k$ categories, $p(y_i = c_t|x_i)$ could take on any value.

The standard practice for computing this loss has been to use "ground truth" labels (in the form of "one-hot" vectors) provided in common benchmark datasets, for example, `ILSVRC15` (Russakovsky et al., 2015), and `CIFAR10` (Krizhevsky, 2009), to train and evaluate models. The single "true" category for each image is decided through human consensus (taking the mode of the distribution over images), or by the database creators. However, this approximation introduces a bias into the learning paradigm that has important distributional implications. Instead of a particular image——and stimulus vector——being associated with a probability mass function over labels, all probability mass is reallocated to the modal category. This forces the network to learn that all instances of a category are equally likely, and discards information about how likely it is to come from others.

The impact of focusing only on the mode can be seen through a simple Bayesian analysis. Assume each category $c$ is associated with a distribution over images $p(x|c)$. Given an image $x_i$, the probability that its label $y_i$ should be $c$ is $p(c|x_i) \propto p(x_i|c)p(c)$. Assuming the prior probabilities of categories do not vary wildly, the only way for $p(c|x_i)$ to be 1 is if $p(x_i|c)$ is infinitely greater than $p(x_i|c')$ for all $c' \neq c$. This removes all supervisory information about the similarity of $c$ to other classes, and potentially places an artificial bound on the expected risk.

Under what circumstances is this a reasonable assumption or approximation? In some extreme cases, where $p(x|c)$ is non-overlapping, such an assumption is justified. In the majority of problems,

however, $p(c|x)$ is clearly not 1 for one particular category, and 0 for all others; nor are all images $x$ equally likely to be associated with $c$ (for example, some images of dogs are more likely than others). Without this information, our classifier will not know that the penalty for mistaking a dog for a cat should be less than mistaking a dog for a car. More importantly, in domains where feature overlap between categories is common (for example, dogs and cats share many joint features), we are forcing our classifier to solve the wrong problem. In order to satisfy the constraint of wholly non-confusable category exemplars, our networks may resort to memorization (which notably satisfies this condition given any dataset with non-identical exemplars across categories).

How, then, can we reach a more natural approximation of $p(y|x)$? For some problems, it is easy to just sample from some real set of data $p(x, y)$. But, for image classification, we must rely on humans as being the gold standard for estimating labels. If we expect human image labels $p_{hum}(c|x)$ to reflect the natural distribution over categories given an image, we can use these samples to define a new expected loss as follows:

$$\int \mathcal{L}(f_\theta, x, y) \, p(x, y) \, dx \, dy \approx \frac{1}{n} \sum_{i=1}^{n} \sum_{c} \mathcal{L}(f_\theta, x_i, c) \, p_{hum}(c|x). \tag{5}$$

In the case where $f_\theta(x)$ is a distribution $p_\theta(y|x)$ and $\mathcal{L}(f, x, y)$ is the log-likelihood $\log p(y|x)$ (or equivalently, cross-entropy to a one-hot vector), the expected loss reduces to the cross-entropy between the human distribution and that predicted by the classifier:

$$\frac{1}{n} \sum_{i=1}^{n} \sum_{c} p_{hum}(c|x_i) \log p_\theta(y_i = c|x_i). \tag{6}$$

This approach naturally complements those seeking to improve generalization by using distributional assumptions around $p(x|y)$ to augment training sets (Zhang et al., 2017)—the human label distribution acts similarly to a kernel on $y$ that distributes mass over neighboring values, although in a way that is potentially also informative about $x$.

## 3 Selection of an Image Dataset

As a first test of the impact of incorporating human uncertainty into image classification, we chose to use the `CIFAR10` image dataset (Krizhevsky, 2009). This dataset played a significant role in the early development of CNNs for image classification. While it has subsequently been replaced by datasets that are larger and higher resolution such as the ImageNet Large Scale Visual Recognition Challenge (Russakovsky et al., 2015), `CIFAR10` has a number of features that make it attractive for testing these ideas.

First, the dataset is relatively small. This makes it possible for us to exhaustively collect substantial amounts of human data on a sizable subset of the dataset. At the same time, it contains a significant number of images of 10 different categories, making it a meaningful test of the approach.

Second, the low resolution of the images is actually useful for producing variation in human responses. Human error rates for high resolution images are sufficiently low that is hard to get a meaningful signal from the responses. Using low-resolution images increases the error rate, and consequently increases the number of human responses that differ from the modal value.

Finally, `CIFAR10` contains a number of examples that are close to the category boundaries, in contrast with other datasets that are more carefully curated such that each image is selected to be a good example of the category. Likewise, subsets of the categories themselves are relatively closely related (e.g., cat, dog, deer, and horse), potentially making human uncertainty informative.

## 4 Dataset Construction

Human judgments were collected for all 10,000 $32 \times 32$ color images in the *testing* subset of `CIFAR10`, which contains 1,000 images for each of the following ten categories: airplane, automobile, bird, cat, deer, dog, frog, horse, ship, truck. This allows us to evaluate models using the same large and well-known test set in terms of a different set of targets. The remaining subset of the data comprises the standard training set of 50,000 images (5,000 per category).

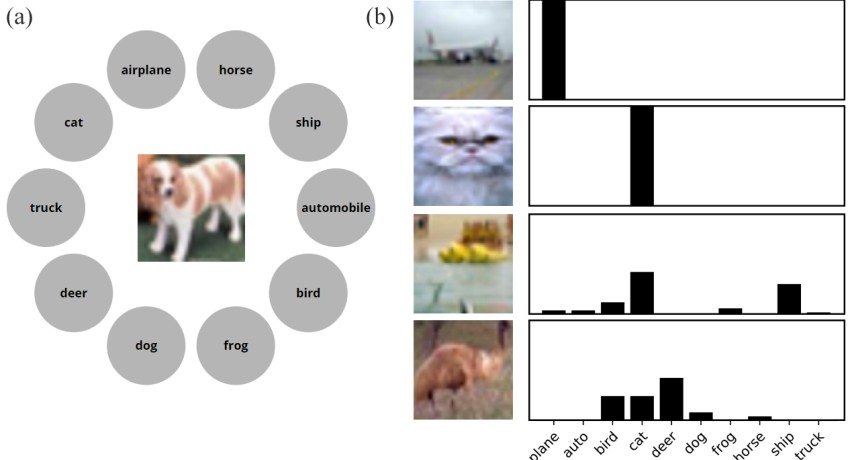

Figure 1: Estimation of $p(c|x)$ via humans. **a.** Experiment web interface for our human categorization task. Participants categorized each image from an order-randomized circular array of the `CIFAR10` labels. **b.** Examples of images and their human choice proportions. For many images (upper plane and cat), choices are unambiguous, matching the `CIFAR10` labels. For others (lower boat and bird), humans are far less certain, even without time constraints.

# 5  HUMAN EXPERIMENTS

Our `CIFAR10H` behavioral dataset consists of 511,400 human categorization decisions (approximately 50 judgments per image) made over our stimulus set collected via Amazon Mechanical Turk (Buhrmester et al., 2011)—to our knowledge, the largest reported in a single study to date.

In the experiment (illustrated in Figure 1a), participants saw an image, presented centrally, and were asked to categorize it by pressing one of the ten labels surrounding the image as quickly and accurately as possible (but with no time limit). Label positions were shuffled between candidates. There was an initial training phase, during which candidates had to score at least $75\%$ accuracy, split into 3 blocks of 20 images taken from the `CIFAR10` training set (6 per category, total). If a candidate failed any block they were asked to redo it until passing the threshold accuracy. For the main experiment, each participant (2,571 total) categorized 200 images, 20 from each category. Every 20 trials there was an attention check image—a carefully selected unambiguous member of a particular category. Participants who scored below $75\%$ on these checks were removed from the final analysis (only 14 participants failed this test).

The mean number of judgments per image was 51 (range: $47 - 63$). The mean accuracy per subject was $95\%$ (range: $71\% - 100\%$). The mean accuracy per image was $95\%$ (range: $0\% - 100\%$). Average completion time was 15 minutes, and workers were paid \$1.50 total. Examples of distributions over categorization judgments for a selection of images is shown in Figure 1b.

# 6  BENCHMARKS

We demonstrate the effectiveness of our dataset as a generalization benchmark by evaluating a range of pretrained SOTA-level CNNs on it both before and after fine-tuning with a subset of our human data (training on 90%, validating on 10%). As a control, we also fine-tune and validate using the corresponding splits of the original `CIFAR10` one-hot labels. Finally, we test all models on the `CIFAR10` training set to assess potential overfitting to fine-tuning sets, as well as both `CIFAR10.1 v4` and `CIFAR10.1 v6`, sets of 2,000-image extensions of `CIFAR10` designed to test the generalization limits of traditionally-trained CNNs using standard one-hot labels (Recht et al., 2018).

Table 1: Crossentropy and accuracy (parentheses) for each validation set and the original `CIFAR10` training set ("c10 50k"). Crossentropy for our human labels decreases substantially after fine-tuning, especially when using human targets. Fine-tuning on human targets also produces the best generalization in terms crossentropy on `CIFAR10.1`, whereas using ground truth (modal) labels has a slight edge in terms of top-1 accuracy.

| Pretrained on `CIFAR10` | | | | | |
|---|---|---|---|---|---|
| | c10h val. | c10 val. | c10.1 v4 | c10.1 v6 | c10 50k |
| `vgg` | 0.78 (94%) | 0.30 (95%) | 0.84 (83%) | 0.79 (85%) | 0.00 (100%) |
| `densenet` | 0.61 (96%) | 0.15 (96%) | 0.51 (88%) | 0.54 (89%) | 0.00 (100%) |
| `pyramidnet` | 0.58 (97%) | 0.14 (97%) | 0.47 (89%) | 0.47 (90%) | 0.00 (100%) |
| `resnet` | 0.78 (94%) | 0.29 (94%) | 0.66 (85%) | 0.74 (85%) | 0.00 (100%) |
| `wrn` | 0.44 (96%) | 0.15 (96%) | 0.40 (90%) | 0.38 (91%) | 0.00 (100%) |
| `wrn_co` | 0.47 (96%) | 0.12 (97%) | 0.38 (90%) | 0.38 (90%) | 0.00 (100%) |
| `rn_preact` | 0.72 (94%) | 0.19 (95%) | 0.59 (87%) | 0.64 (86%) | 0.00 (100%) |
| `shake` | 0.60 (98%) | 0.09 (98%) | 0.34 (92%) | 0.33 (92%) | 0.00 (100%) |

| Fine-tuned on `CIFAR10` | | | | | |
|---|---|---|---|---|---|
| | c10h val. | c10 val. | c10.1 v4 | c10.1 v6 | c10 50k |
| `vgg` | 0.50 (93%) | 0.21 (93%) | 0.66 (82%) | 0.58 (84%) | 0.03 (99%) |
| `densenet` | 0.59 (96%) | 0.14 (96%) | 0.47 (88%) | 0.50 (89%) | 0.00 (100%) |
| `pyramidnet` | 0.50 (97%) | 0.11 (98%) | 0.37 (90%) | 0.38 (91%) | 0.00 (100%) |
| `resnet` | 0.61 (93%) | 0.24 (93%) | 0.62 (83%) | 0.63 (84%) | 0.04 (99%) |
| `wrn` | 0.38 (96%) | 0.12 (96%) | 0.38 (89%) | 0.33 (91%) | 0.00 (100%) |
| `wrn_co` | 0.44 (97%) | 0.11 (97%) | 0.36 (90%) | 0.36 (91%) | 0.00 (100%) |
| `rn_preact` | 0.65 (94%) | 0.17 (94%) | 0.56 (87%) | 0.56 (87%) | 0.00 (100%) |
| `shake` | 0.50 (98%) | 0.07 (98%) | 0.28 (92%) | 0.28 (93%) | 0.00 (100%) |

| Fine-tuned on `CIFAR10H` | | | | | |
|---|---|---|---|---|---|
| | c10h val. | c10 val. | c10.1 v4 | c10.1 v6 | c10 50k |
| `vgg` | 0.35 (93%) | 0.21 (93%) | 0.55 (82%) | 0.48 (85%) | 0.07 (99%) |
| `densenet` | 0.32 (95%) | 0.17 (95%) | 0.40 (87%) | 0.38 (88%) | 0.09 (98%) |
| `pyramidnet` | 0.28 (97%) | 0.11 (97%) | 0.32 (89%) | 0.30 (90%) | 0.04 (100%) |
| `resnet` | 0.36 (92%) | 0.24 (92%) | 0.56 (82%) | 0.54 (82%) | 0.23 (93%) |
| `wrn` | 0.27 (97%) | 0.12 (97%) | 0.32 (90%) | 0.29 (91%) | 0.03 (100%) |
| `wrn_co` | 0.28 (96%) | 0.13 (96%) | 0.37 (88%) | 0.34 (89%) | 0.06 (100%) |
| `rn_preact` | 0.33 (94%) | 0.18 (94%) | 0.45 (85%) | 0.43 (86%) | 0.08 (99%) |
| `shake` | 0.26 (97%) | 0.10 (98%) | 0.28 (91%) | 0.27 (91%) | 0.04 (100%) |

## 6.1 PRETRAINED MODELS

We pretrained seven CNN architectures (and eight models total) on `CIFAR10` (Table 1), including a number of `CIFAR10` benchmark breakers over the last 5 years, as well as the current state-of-the-art model, shake-shake with cutout (Gastaldi, 2017; Devries & Taylor, 2017):

- VGG: `vgg_15_BN_64`
- DenseNet: `densenet_BC_100_12`
- PyramidNet: `pyramidnet_basic_110_270`
- ResNet: `resnet_basic_110`
- Wide ResNet: `wrn_28_10`, `wrn_28_10_cutout16`
- ResNet Pre-act: `resnet_preact_bottleneck_164`
- Shake Shake: `shake_shake_26_2x64d_SSI_cutout16`

All models were trained using PyTorch (Paszke et al., 2017), using the repository found at `https://github.com/hysts/pytorch_image_classification`. We used the de-

Table 2: Crossentropy and accuracy (parentheses) for each validation set when combining *mixup* with original or human targets.

| Pretrained on `CIFAR10` | | | | | |
|---|---|---|---|---|---|
| | c10h val. | c10 val. | c10.1 v4 | c10.1 v6 | c10 50k |
| `wrn_28_10_co16` | .45 (96%) | .14 (96%) | .37 (90%) | .38 (90%) | .00 (100%) |
| `shake_26_2x64d_SSI_co16` | .55 (97%) | .09 (97%) | .34 (92%) | .33 (92%) | .00 (100%) |

| Fine-tuned on `CIFAR10` with *mixup* | | | | | |
|---|---|---|---|---|---|
| | c10h val. | c10 val. | c10.1 v4 | c10.1 v6 | c10 50k |
| `wrn_28_10_co16` | .34 (96%) | .12 (96%) | .37 (89%) | .33 (90%) | .03 (100%) |
| `shake_26_2x64d_SSI_co16` | .55 (97%) | .09 (97%) | .34 (92%) | .33 (92%) | .00 (100%) |

| Fine-tuned on `CIFAR10H` with *mixup* | | | | | |
|---|---|---|---|---|---|
| | c10h val. | c10 val. | c10.1 v4 | c10.1 v6 | c10 50k |
| `wrn_28_10_co16` | .28 (96%) | .14 (96%) | .36 (89%) | .31 (90%) | .07 (100%) |
| `shake_26_2x64d_SSI_co16` | .26 (97%) | .12 (97%) | .30 (91%) | .28 (91%) | .08 (99%) |

fault hyperparameters in the repository for all models, following (Recht et al., 2018) for the sake of reproducibility.

## 6.2 MODEL FINE-TUNING

We fine-tuned each pretrained model using our human data. As opposed to traditional one-hot vectors, our targets were parameter vectors reflecting the aggregate human classifications for each image, normalized to sum to one. We used 9000 image-label pairs from our `CIFAR10H` as our training set, and held out a set of 1000 image-label pairs for validation. As a control, we also trained using the original `CIFAR10` labels for the 9000 images in our training set, and 1000 images in our validation set. We trained models for 100 epochs using *adam* (Kingma & Ba, 2014) on the loss between model predictions and human labels, and performed a grid-search over learning rates (0.1, 0.01, and 0.001), and random seed (3 values).

## 6.3 EVALUATION MEASURES

In order to examine the effects of tuning CNN classification models on our dataset, we assessed their performance on our validation and test sets before, during, and after training (Table 1 shows the best of these scores). We choose crossentropy as our evaluation metric of choice given that our human targets are meant to capture probabilities for each class given an image. All validation and test sets are described in detail below.

- c10h val. — 1000-image validation for our `CIFAR10H`
- c10 val. — 1000-image validation for our control using `CIFAR10`
- c10.1 v4 — 2000-image test set (Recht et al., 2018)
- c10.1 v6 — 2000-image test set (alternative; Recht et al. (2018))
- c10 50k — Original 50k-image `CIFAR10` training set

## 7 RESULTS

### 7.1 PERFORMANCE OF PRETRAINED MODELS ON BENCHMARK

Our first finding is that the loss (crossentropy) for models pretrained on `CIFAR10` increases substantially when they are evaluated on our benchmark, `CIFAR10H` (Table 1, upper section, columns

1 and 2). Accuracy, by contrast, remains roughly the same. This supports the findings of Recht et al. (2018), who found that pretrained `CIFAR10` CNNs generalize poorly even to validation sets sampled as closely as possible to the original. We replicate these findings in columns 3 and 4 of the same table.

## 7.2 TRAINING (FINE-TUNING) ON `CIFAR10H`

Our main finding is that models fine-tuned on our set of human labels exhibit generalization performance—in terms of loss—that is consistently better than our control (Table 1, section 2). These effects were much more pronounced that those seen with our control training paradigm, in which we further trained / fine-tuned pretrained networks with the ground-truth labels originally associated with CIFAR10 test images (Table 1). We can visualize these effects more clearly by examining individual runs for models, with a representative example shown in Figure 3.

Here, we trained to the full distribution of human guesses for each image, rather than sampling one-hot labels from the distribution as training input. These paradigms are equivalent mathematically, as the CNN softmax itself is probabilistic; however, training to samples instead of the full probability vector takes much longer to train, as it may take some time before multiple modes are sampled.

## 7.3 COMBINING HUMAN LABEL UNCERTAINTY WITH *mixup*

Approaching the problem of extending empirical risk minimization paradigms to avoid training and validation set overfitting in a complementary framework, Zhang et al. and colleagues, 2017, introduced a technique called *mixup*, in which datasets are augmented by including convex combinations of image vectors and their labels. This can be thought of as estimating the true distribution $p(x, y)$ by using a joint kernel over labels and images, with the result that classification model is regularized to favor simple linear behavior in-between training examples. As adding *mixup* to several CNN models on `CIFAR10` improves their test set error and robustness to adversarial attacks, we assessed the performance of using *mixup* instead of and alongside our labels.

Results are presented in Table 2. We find that using *mixup* decreases loss, both on validation and generalization datasets, but, importantly, that combining *mixup* with our data decreases this loss further. This is as expected, considering our dataset supplies richer information about labels, with which the out-of-sample linearizing effects of *mixup* can utilize.

## 7.4 IMPLICATIONS FOR ADVERSARIAL DEFENSE

Because our human targets contain information about images near perceptual boundaries, we might expect that representations learned in service of predicting them would be more robust to adversarial attacks, particularly in cases where similar categories make for good attack targets. We generally measure this by looking again at maximized crossentropy after a basic Fast Gradient Sign Method (FGSM) attack. Results using two of the best models from architecture classes randomly selected from our total set are presented in Table 3 (no other models were tested at any point). As we might expect, accuracy after an attack on the entire `CIFAR10` test dataset is lower when training on human targets, because the top-1 choice becomes less confident, however, crossentropy is notably much lower when using networks supervised to human targets, indicating a trade-off between accuracy and robustness.

## 8 DISCUSSION

We have shown that incorporating human uncertainty about the categories that apply to images into the objective function used for training image classification systems can result in an improvement in both generalization and robustness. In the remainder of the paper we will briefly discuss some of the limitations and implications of these results.

We focused on the `CIFAR10` dataset in part because its manageable size made it was possible to obtain human judgments for a reasonable subset of images. Even so, our human dataset covers only a sixth of the complete dataset. Getting human judgments for any modern dataset in full is potentially too costly, which places limits on the practical benefits of this approach, but the results

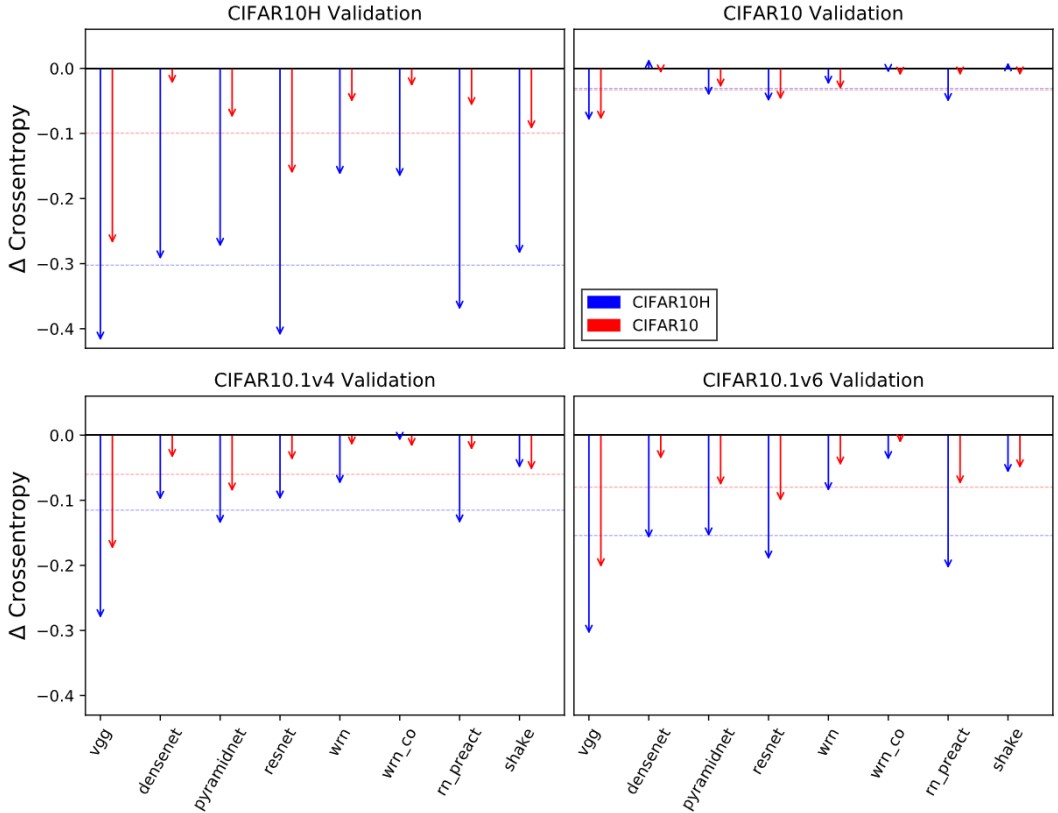

Figure 2: Change in loss ($\Delta\mathcal{L}$) for human targets, original `CIFAR10` targets, and `CIFAR10.1` targets after fine-tuning using original `CIFAR10` targets (red) or `CIFAR10H` targets (blue). Average $\Delta\mathcal{L}$ across models for each training condition is represented by a dashed horizontal line.

Table 3: FGSM attacks on the `CIFAR10`-tuned and `CIFAR10H`-tuned networks. Using human labels results in lower crossentropy (lower is better), but also lower accuracy (parentheses).

| Fine-tuned with `CIFAR10` | | |
|---|---|---|
| | Before FGSM | After FGSM |
| `pyramidnet` | .08 (98%) | 5.22 (22%) |
| `shake_shake` | .05 (98%) | 4.18 (39%) |

| Fine-tuned with `CIFAR10H` | | |
|---|---|---|
| | Before FGSM | After FGSM |
| `pyramidnet` | .07 (99%) | 3.52 (19%) |
| `shake_shake` | .06 (99%) | 2.10 (38%) |

we present here are sufficient to demonstrate that human uncertainty carries important information about the similarity structure of images and categories, and that it can be used as an evaluation where training itself is not feasible.

To extend these results to a larger scale, it might be possible to build predictive models that extend the patterns of uncertainty observed in our data to larger datasets. For example, taking a softmax over a semantic distance measure on category labels might give enough of this type of structure to improve training. This is essentially a way of constructing a probabilistic kernel on category labels, complementary to strategies aimed at countering overfitting on image datasets by augmenting them via sampling extra training inputs using domain-specific alterations—such as slight image rotations

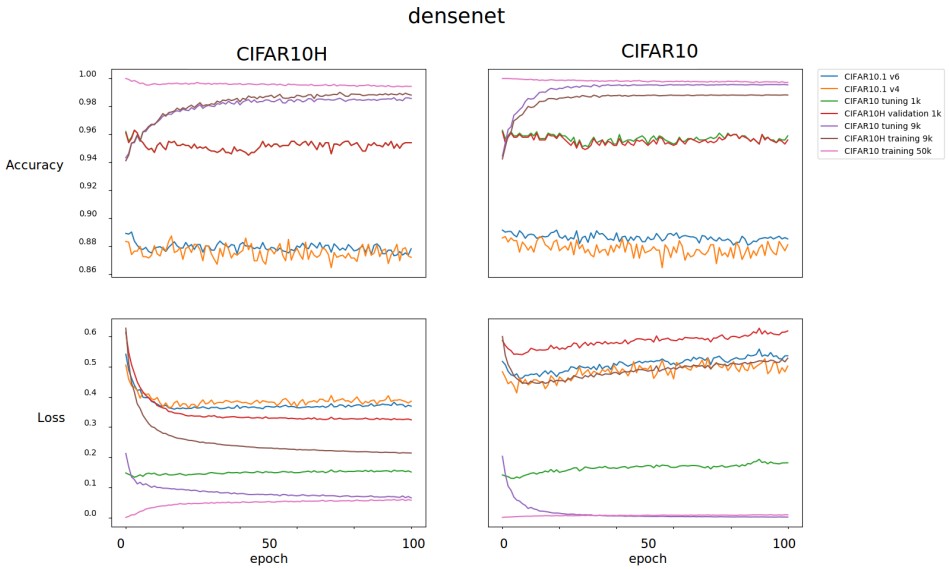

Figure 3: Representative fine-tuning results for densenet, on our data (left) and original cifar labels (right). Results are shown for a learning rate of 0.001, and with a seed of 0.

and horizontal reflections—or distributional assumptions over the perceptual space surrounding empirical examples (Schott et al., 2018; Simonyan & Zisserman, 2014).

A second issue with applying this approach to other datasets is that higher-resolution and better-curated collections of images are less likely to produce human uncertainty. In some ways, careful selection of good examples of particular categories may undermine the goal of producing robust image classification systems, because it means that there are few images that are informative about category boundaries. Having more confusable images, together with the information about exactly how humans find them confusing, may thus be a way to improve generalization and robustness more broadly.

Capturing the relationships that exist between categories may also be important in thinking about the robustness of systems to adversarial examples. For instance, knowing that dogs and cats are similar means that we should be less concerned when it is easy to make an image classification system confuse one for the other. However, dogs and cars are very different, and we should hope that driverless cars are sensitive to that difference.

Human beings remain the best examples we have of systems that are capable of solving certain problems, including classifying images in a way that is generalizable and robust. Collecting more information about how people do this can only improve our machine learning systems. In particular, the patterns of errors that people make and the uncertainty of their display are informative about the structure of the world around us, and provide an additional source of information that machine learning systems can exploit.

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
