# OpenReview forum: "Improving machine classification using human uncertainty measurements"
_ICLR.cc/2019/Conference_

### Official Review · AnonReviewer2 · 2018-11-02

**Rating:** 3
**Confidence:** 2

**Review:**

The paper presents a new version of CIFAR10 that is labelled by multiple people (the test part of the data). They use it to improve the calibration of several image classifiers through “fine-tuning” and other techniques
The title is too general, taking into account that this setting has appeared in classification in many domains, with different names (learning from class distributions, crowd labellers, learning from class scores, etc.). See for instance,
https://www.ncbi.nlm.nih.gov/pmc/articles/PMC3994863/
http://www.cs.utexas.edu/~atn/nguyen-hcomp15.pdf
Also, at the end of section 2 we simply reach logloss, which is a traditional way of evaluating the calibration of a classifier, but other options exist, such as the Brier score. At times, the authors mention the trade-off between classification accuracy and cross-entropy. This sounds very much the trade-off between refinement and calibration, as one of the possible decompositions of the Brier score.
The authors highlight the limitations of this work, and they usually mention that the problem must be difficult (e.g., low resolution). Otherwise, humans are too good to be useful. I suggest the authors to compare with psychophysics and possible distortions of the images, or time limits for doing the classifications.
Nevertheless, the paper is not well motivated, and the key procedures, such as “fine-tuning” lack detail, and comparison with other options.
In section 2, which is generally good and straightforward, we find that p(x|c) being non-overlapping as a situation where uncertainty would be not justified. Overlap would simply say that it is a categorisation (multilabel classification) problem rather than a classification problem, but this is different from the situation where labels are soft or given by several users.
In the end, the paper is presented from the perspective of image recognition, but it should be compared with many other areas in classification evaluation where different metrics, presentation of the data, levels of uncertainty, etc., are used, including different calibration methods, as alternatives to the expensive method presented here based on crowd labelling.
Pros:
-	More information about borderline cases may be useful for learning. This new dataset seems to capture this information.
Cons:
-	The extra labelling is very costly, as the authors recognise.
-	The task is known in the classification literature, and a proper comparison with other approaches is required.
-	Not compared with calibration approaches or other ways where boundaries can be softened with less information from human experts. For instance, a cost matrix about how critical a misclassification is considered by humans (cat <-> dog, versus cat <-> car) could also be very useful, and much easier to obtain.

---

### Official Review · AnonReviewer1 · 2018-11-02
**Multiple GT labels**

**Rating:** 3
**Confidence:** 5

**Review:**

The authors propose to improve classification accuracy in a supervised learning framework, by providing richer ground truth in the form a distribution over labels, that is not a Dirac delta function of the label space. This idea is sound and should improve performance.

Unfortunately this work lacks novelty and isn't clearly presented.
(1) Throughout the paper, there are turns that used without definition prior to use, all table headers in table 1.
(2) Results are hard to interpret in the tables, and there are limited details. Mixup for example, doesn't provide exact parameters, but only mentions that its a convex sum.
(3) There is no theoretical justification for the approach.
(4) This approach isn't scalable past small datasets, which the authors acknowledge.
(6) This has been already done. In the discussion the authors bring up two potential directions of work:
   (a) providing a distribution over classes by another model - > this is distillation (https://arxiv.org/abs/1503.02531)
   (b) adding a source of relationships between classes into the objective function -> this is (https://static.googleusercontent.com/media/research.google.com/en//pubs/archive/42854.pdf)

---

### Official Review · AnonReviewer3 · 2018-11-02
**Interesting idea, but the empirical investigation seems lacking**

**Rating:** 6
**Confidence:** 4

**Review:**

The authors create a new dataset with label distributions (rather than one-hot annotations) for the CIFAR-10 test set. They then study the effect of fine-tuning using this dataset on the generalization performance of SOTA deep networks. They also study the effects on adversarial robustness.

I think that datasets such as the one generated in this paper could indeed be a valuable testbed to study deep network generalization and robustness. There are many nice benefits of label distributions over one hot labels (that the authors summarize in Section 2.) The paper is also clear and well-written.

That being said, I do not find the investigation of this paper completely satisfactory. For instance in the generalization experiments, the numbers presented seem to show some interesting (and somewhat surprising) trends, however the authors do not really pursue these or provide any insight as to why this is the case. I also find the section on robustness very weak.

Detailed comments:

- The theoretical contribution mentioned in the appendix does not really seem to be a contribution - it is just a simple derivation of the loss under label distributions. Theoretical contributions are not necessary for a paper to have merit - the authors should remove this statement from the introduction as it detracts from the value of the paper.

- I find it somewhat surprising that the accuracy of the models does not change on training with Cifar10H. Do the authors have any intuition as to why this is the case? The model cross entropy seems to go down, indicating that probability assigned to the correct class increases. I would think that training with label distributions would actually reduce misclassification on confusing instances. It would be interesting to see how the logit distributions change for different examples. For instance, how does the model confidence change on correctly vs wrongly classified examples?

- The authors mention that they run each hyperparameter configuration for three random seeds. It would be nice then to see error bars for the results reported Tables 1 and 2, particularly because the differences in accuracy are small. Did the authors try different train-test splits of the test set? It would also be helpful if the authors could make plots for the results in these tables (at least in the appendix). It is hard to compare numbers across different tables.

-I find the results in Table 2 confusing. Comparing the numbers to Table 1, it seems that mixup does not really change accuracies/loss. The model names in Table 2 do not exactly match Table 1 so it is hard to identify the additional gain from using mixup that the authors mention. The authors should add plots for these results to illustrate the effect of adding mixup more clearly.

-I am not convinced by the section on robustness. Firstly, it is not clear to me why the authors chose FGSM which is known to be a somewhat simple attack to illustrate improved robustness of their model. To perform a useful study of robustness, the authors should study SOTA attacks such as PGD [Madry et al., 2017]. I also do not understand the claim that the top-1 choice becomes less confident after training with CIFAR10H -- this seems to be contradicted by the fact that the cross entropy loss goes down. The authors should provide supporting evidence for this claim by looking at changes in confidence (see point 3 above). Also, the comment about the trade-off between accuracy and robustness seems vague - could the authors clarify what they mean?

Overall, I like the premise of this paper and agree that with the potential benefits of the dataset generated. However, I think that the current experiments are not strong enough to corroborate this.

---

### Public Comment · (anonymous) · 2018-10-04
**Interesting data**

It is a neat idea to, for CIFAR classification problem, create soft labels by collecting human supervision, which should be useful to improve generalization ability. Do you have any plan to release the data CIFAR10H?

---

> ### Author Response · Authors · 2018-10-04
> **Dataset release**
>
> Thanks for your interest. We are indeed planning to release the dataset (both the aggregate and individual human responses) as soon as the paper lands. We also intend to release some code and trained models. It's intended to serve as a new benchmark, target of research questions, and even a training/tuning dataset.

---

> > ### Public Comment · (anonymous) · 2018-10-09
> > **That's nice**
> >
> > Thanks. I would like to use your data/code and I'm waiting for the release.

---

### Meta-Review · Area_Chair1 · 2018-12-13

**Confidence:** 5
**Recommendation:** Reject

**Metareview:**

 The paper presents a new annotation of the CIFAR-10 dataset (the test set) as a distribution over labels as opposed to one-hot annotations. This datasets forms a testbed analysis for assessing the generalization abilities of the state-of-the-art models and their robustness to adversarial attacks.

All the reviewers and AC acknowledge the contribution of dataset annotation and that the idea of using label distribution for training the models is sound and should improve the generalization performance of the models.
However the reviewers and AC note the following potential weaknesses: (1) the paper requires major improvement in presentation clarity and in-depth investigation and evidence of the benefits of the proposed framework – see detailed comments of R3 on what to address in a subsequent revision; see the suggestions of R2 for improving the scope of the empirical evaluations (e.g. distortions of the images, incorporating time limits for doing the classifications) and the requests of R1 for clarifications; (2) the related work is inadequate and should be substantially extended – see the related references suggested by the R2; also R1 rightly pointed out that two out of four future extensions of this framework have been addressed already, which questions the significance of findings in this submission.
The R2 raised concerns that the current evaluation is missing comparisons to a) the calibration approaches and b) cheaper/easier ways of getting soft labels -- see R2’s suggestion to use the Brier score for model calibration and to use a cost matrix about how critical a misclassification is (cat <-> dog, versus cat <-> car) as soft labels.
Among these, (2) did not have a substantial impact on the decision, but would be helpful to address in a subsequent revision. However, (1) and (3) makes it very difficult to assess the benefits of the proposed approach, and was viewed by the AC as a critical issue.

There is no author response for this paper. The reviewer with a positive view on the manuscript (R3) was reluctant to champion the paper as the authors have not addressed the concerns of the other reviewers (no rebuttal).